# Improvement of Corrosion and Wear Resistance of CoCrNiSi_0.3_ Medium-Entropy Alloy by Sputtering CrN Film

**DOI:** 10.3390/ma16041482

**Published:** 2023-02-10

**Authors:** Yi-Chun Chang, Kaifan Lin, Ju-Lung Ma, Han-Fu Huang, Shih-Hsien Chang, Hsin-Chih Lin

**Affiliations:** 1Department of Materials Science and Engineering, National Taiwan University, Taipei 10617, Taiwan; 2Department of Materials and Mineral Resources Engineering, National Taipei University of Technology, Taipei 10608, Taiwan

**Keywords:** medium-entropy alloy, magnetron sputtering, CrN thin film, wear, corrosion

## Abstract

In this study, Co, Cr, and Ni were selected as the equal-atomic medium entropy alloy (MEA) systems, and Si was added to form CoCrNiSi_0.3_ MEA. In order to further improve its wear and corrosion properties, CrN film was sputtered on the surface. In addition, to enhance the adhesion between the soft CoCrNiSi_0.3_ substrate and the super-hard CrN film, a Cr buffer layer was pre-sputtered on the CoCrNiSi_0.3_ substrate. The experimental results show that the CrN film exhibits a columnar grain structure, and the film growth rate is about 2.022 μm/h. With the increase of sputtering time, the increase in CrN film thickness, and the refinement of columnar grains, the wear and corrosion resistance improves. Among all CoCrNiSi_0.3_ MEAs without and with CrN films prepared in this study, the CoCrNiSi_0.3_ MEA with 3 h-sputtered CrN film has the lowest wear rate of 2.249 × 10^−5^ mm^3^·m^−1^·N^−1^, and the best corrosion resistance of I_corr_ 19.37 μA·cm^–2^ and R_p_ 705.85 Ω·cm^2^.

## 1. Introduction

High entropy alloys (HEAs) consisting of multi-principal elements are changing the traditional definition of metallurgical alloys and exploring the unknown territories of alloy design [1]. Recent reports illustrate the excellent mechanical properties of HEAs in many aspects, such as the trade-off between tensile strength and ductility at cryogenic temperatures [2] and acceptable mechanical endurance in high-temperature environments [2,3]. Several studies attempt to elevate the mechanical performance of novel HEA systems through solid solution strengthening [4,5], precipitation hardening [6,7], and grain refinement strengthening [8]. Although the multi-principal element alloy has brought unexpected mechanical improvement, a small amount of impurities in the complicated lattice matrix also result in unpredictable harassment. Due to the limitations of the purity of raw manganese metal, the sulfur-containing inclusions that served as pitting initiation sites weaken the corrosion resistance of CoCrFeMnNi HEA. Moreover, it has been reported in the literature that impurities forming Mn-Cr-Al oxides surrounded by Mn(S, Se) shells may significantly reduce the mechanical properties [9]. The reduction of Mn-related inclusions and the increase of passivation-induced Cr_2_O_3_ content in CoCrFeMn_x_Ni HEAs result in better corrosion resistance [10]. Therefore, to improve the mechanical properties or corrosion resistance, it is reasonable to reduce the content of Fe and Mn raw materials, which may contain more impurities.

Along with the development of high-entropy alloys, researchers have also focused on medium-entropy alloys, such as the equi-atomic CoCrNi series of alloys that exhibit an excellent strength-ductility trade-off [11]. Fundamental understanding of CoCrNi MEAs shows their development potential [12], especially the exceptional mechanical properties in the cryogenic environment [13,14] and impressive performance in wear and corrosion resistance [15,16]. It has been reported that the silicon addition in the high entropy alloy can improve the mechanical properties and corrosion resistance [17,18]. The corrosion resistance of CoCrFeMnNi HEA has been improved by replacing the unstable MnO·Cr_2_O_3_ inclusions in the Cr depletion zone with Si-induced MnO-SiO_2_ [17]. Recently, the development trend of CoCrNi MEAs also tends to introduce silicon into the current alloy system, which leads to a quantitative improvement in mechanical strength [19]. 

The wear resistance of HEA alloys (especially with FCC structure) is not satisfactory due to their low surface hardness, and their wear resistance has been improved by methods such as boriding, nitriding, and carburizing [20,21,22,23]. The nitriding process will produce various nano-precipitates of related chromium nitride to form a protective layer on the surface of Al_x_CoCrFeNi HEA, and such a nitriding layer will also significantly increase the surface hardness [24]. However, the nitriding or boronizing protective layer cannot be fully uniform, which leads to the instability of the macroscopic performance. In addition, to obtain a uniform and stable protective layer, sputtering technology offers a fast deposition rate and a low-cost solution. Although the development of HEA systems has been going on for more than a decade, there are few reports on the deposition of stable protective films on HEA surfaces by conventional reactive sputtering techniques. Many studies have shown that sputter deposition of CrN film on conventional stainless steel is beneficial for its wear and corrosion resistance [25,26]. The CrN film deposited on the surface of 316 stainless steel has a positive effect on the friction behavior in NaCl solution and high temperature environments until heated to 550 °C [27]. In the tribo-corrosion test, the CrN coating layer can also remain relatively stable during the ball-on-plate sliding under various applied potentials [28]. As stainless steels are similar to CoCrNi-based MEAs in terms of their single FCC structure and lattice constant, this study attempts to sputter CrN film with a Cr buffer layer on the surface of CoCrNiSi_0.3_ MEA and discusses in detail the improvement of corrosion and wear resistance of CoCrNiSi_0.3_ MEA by sputtering CrN film.

## 2. Materials and Methods

### 2.1. Materials

The Si-added, equal-atomic CoCrNi alloy system, CoCrNiSi_0.3_ (in at. %) MEA, was selected as the substrate for CrN deposition in this study. The purity of Co, Cr, Ni, and Si raw materials was higher than 99.99 wt.%, and the ingot was prepared in a vacuum arc remelting furnace (VAR). The ingot was repeatedly melted more than five times by turning it over to ensure its uniform composition. A series of thermomechanical treatments were carried out to further improve the properties, as shown in Figure 1. The ingot was homogenized at 1100 °C for five hours and then hot-rolled at 1100 °C to a thickness of 3.00 mm. Thereafter, the hot-rolled plate was cold-rolled from 3.00 mm to a thickness of 1.00 mm and then cut into test pieces with the size of 20 × 20 × 1 mm^3^. The test pieces were annealed at 900 °C for 15, 30, and 60 min to achieve various grain-size equiaxed grains. The test pieces were mechanically polished and used as the substrate for sputtering CrN films. A conventional direct current (DC) magnetron sputtering system, the LT-PVD400 of Leitai Vacuum Co., Ltd., Taoyuan, Taiwan) with a chromium target (99.99 wt.%), was used for sputtering CrN films. The CrN films were deposited not only on the CoCrNiSi_0.3_ MEA but also on a silicon substrate to successfully prepare the specimen for cross-sectional observation. The sputtering parameters were as follows: the sputtering time of 2, 2.5, and 3 h, the bias voltage of −35 V, the sputtering power of 100 W, the working pressure of 0.8 Pa, the gas flow of 45/30 (Ar/N2) sccm, and the deposition temperature of 250 °C. Since the hardness of CoCrNiSi_0.3_ MEA is about 220 HV, which is quite different from that of CrN (1700~2200 Hv) [29], the huge gap in hardness will lead to poor adhesion between the substrate and the deposited layer and cannot be deposited smoothly [30,31]. Therefore, a thin Cr film is pre-deposited on the surface of CoCrNiSi_0.3_ MEA as a buffer layer. The thickness of the Cr buffer layer is 0.9 μm, and its hardness is reported to be about 8.05 GPa [32]. The sputtering parameters of the Cr buffer layer were set at a working pressure of 0.8 Pa, a bias voltage of −35 V, an Argon input flow rate of 45 sccm, an applied DC power of 100 W, and a sputtering time of 15 min.

### 2.2. Characterization

A field emission scanning electron microscope (JEOL JSM-7800F Prime, Akishima, Japan) with an Oxford Nordlys EBSD detector (Oxford, UK) was used to determine the grain size of the annealed CoCrNiSi_0.3_ alloy. The specimens for EBSD observation were prepared by mechanical grinding, mechanical polishing in an Al_2_O_3_-dilute solution, and then electrochemical polishing in a solution of 90% ethanol and 10% perchloric acid at a voltage of 20 V, which aimed to remove the surface residual stress layers introduced during the mechanical polishing. The surface and cross-sectional morphologies of the CrN films were analyzed by a field emission scanning electron microscope (NOVA NANO SEM 450, Lincoln, NE, USA). The phase identification of the sputtered CrN films was carried out by X-ray diffraction (XRD, Rigaku TTRAX^3^ (Tokyo, Japan), Cu Ka radiation, scan rate: 4°/min, scan step: 0.02°). The hardness and wear resistance of the sputtered CrN films were measured by a micro-Vickers hardness tester (Taiwan Nakazawa Co., Ltd. (Taiwan, China)), and a wear test machine (Freeform P.M. Co., Ltd., London, UK), respectively. According to the ASTM G99 test standard, the wear test was performed against a tungsten carbide ball (hardness: 90HRA) in a pin-on-disk mode under a load of 2 N at a fixed rotation rate of 0.2 m/s and 6000 cycles of 12 mm rotation diameter. Then the friction coefficient, wear volume loss, and wear rate were measured. The formula V = A·d is used to calculate the wear volume losses, where V is the volume of wear loss, A is the wear area, and d is the wear depth. Following the ASTM G99-03 standard, the values of the formulas were calculated through SE Area software. The formula for wear rate is W = V/(S·L), where W is the wear rate, V is the volume of wear loss, S is the sliding distance, and L is the normal load applied. The wear track was observed using a scanning electron microscope (JEOL JSM6510), and the wear track depths were measured by a precision surface roughness meter (KOSAKA SE300, Tokyo, Japan). A three-electrode electrochemical cell was used for the electrochemical test, according to the ASTM G59-97 standard. The tested specimen was set up as a working electrode. The platinum sheet was used as the counter electrode, and the saturated calomel electrode (Ag/AgCl) was the reference electrode. The tested specimen was set up as a working electrode with a reaction area of 2.01 cm^2^. The electrochemical measurement was performed in a 1 M H_2_SO_4_ corrosive solution at room temperature using a potentiostat (Autolab PGSTATM 204). The scanning speed was kept at 0.01 Vs^−1^ from an initial potential of −2.0 V to a final potential at 2.0 V. A Corr-View software was used to obtain the polarization curve and analyze the polarization resistance (R_p_), corrosion potential (E_corr_), and corrosion current density (I_corr_) of CrN/CoCrNiSi_0.3_. Under each heat treatment condition, at least three samples were prepared to test the wear and corrosion properties. A 200 kV field emission transmission electron microscope (FETEM, FEI Tecnai G2 F20) was used to observe the microstructures of CrN thin films. The TEM specimens were prepared by a focused ion beam microscope (FEI Helios 600i). Atomic force microscope analysis (AFM, Park XE-100) provided images with a near-atomic resolution for measuring surface topography by scanning the surface of the test specimens with a tiny probe. It could quantify the surface roughness of specimens down to the angstrom scale.

## 3. Results and Discussion

### 3.1. Microstructure of CoCrNiSi_0.3_ MEA

Figure 2 shows the electron back-scattering diffraction (EBSD) images of the 900 °C-annealed CoCrNiSi_0.3_ MEA. It can be clearly seen that after annealing at 900 °C for 15, 30, and 60 min, the cold-rolled CoCrNiSi_0.3_ MEA is fully recrystallized and exhibits a single-phase FCC structure with grain sizes of 12.6 um, 17.5 um, and 28.2 um, respectively. Meanwhile, plenty of annealed twins are also observed in the grains of all specimens. This indicates the formation of annealed twins is promoted during the annealing [33]. It is obvious that the mechanical properties of CoCrNiSi_0.3_ MEA will be improved due to grain refinement, and the formation of annealing twins will also contribute to a certain extent.

### 3.2. Wear and Corrosion Resistance of CoCrNiSi_0.3_ MEA

Figure 3a shows the plots of friction coefficient and Vickers hardness vs. the annealing time of CoCrNiSi_0.3_ MEA. The data for friction coefficient and hardness are listed in Table 1. The results demonstrate that the friction coefficient gradually increases from 0.981 to 1.367 with the increase in annealing time; while the average hardness gradually decreases from 356 ± 14 to 199 ± 6 Hv. The as-rolled CoCrNiSi_0.3_ MEA has the highest hardness due to the high density of dislocations induced by cold rolling. When the annealing process reaches 15 min, the recrystallization eliminates the internal strain and residual stress stored in the previously deformed specimen, thus reducing the hardness, and the grain-boundary strengthening replaces the deformation-induced strengthening [34]. With the increase in annealing time, the grain size gradually grows, the strengthening effect of the grain boundary weakens, and the hardness continues to gradually decrease again [35]. When the annealing time is 60 min, the hardness reaches its lowest value in this study. Additionally, the friction coefficient decreases with the increase in annealing time because the specimens with higher hardness are more resistant to the wear of the grinding ball. Figure 3b shows the wear volume loss and wear rate of CoCrNiSi_0.3_ MEA as a function of annealing time. The definition of the wear rate is the wear volume loss per unit load and per unit distance. The data for wear volume loss and wear rate are also listed in Table 1. Obviously, the as-rolled specimen has the lowest friction coefficient, the lowest wear volume loss of 1.773 × 10^−1^ mm^3^ and the lowest wear rate of 3.174 × 10^−4^ mm^3^·m^−1^·N^−1^, indicating the best wear performance among the current specimens. However, as the annealing time increases to 60 min, the wear volume loss increases to 2.626 × 10^−1^ mm^3^, and the wear rate increases to 4.896 × 10^−4^ mm^3^·m^−1^·N^−1^. These results indicate that as the annealing time increases, the hardness of the specimens decreases, and thus the wear resistance decreases.

Figure 4 shows the potentiodynamic polarization curves of various CoCrNiSi_0.3_ MEAs tested in a 1 M H_2_SO_4_ solution. The corrosion current density (I_corr_), polarization corrosion potential (E_corr_), and polarization impedance (R_p_) are obtained by Tafel extrapolation, as listed in Table 2. In Figure 4, a passivation behavior occurs for every specimen during the corrosion process. For as-rolled specimens, the high-strain grains and boundaries accompanied by high residual stress are prone to severely corroding [36,37,38,39]. Consequently, the as-rolled specimen has the highest corrosion current density, 470.72 μA·cm^−2^, and the lowest polarization impedance of 98.82 Ω·cm^2^, showing the worst corrosion performance. As mentioned above, the annealing treatment eliminates the internal strain and residual stress, resulting in equiaxed grains instead of as-rolled structures, and hence improves the corrosion performance. In addition, the grain size increases with increasing annealing time, and the effect of grain boundary corrosion is weakened by the reduction of total grain boundaries. Therefore, the corrosion resistance is positively correlated with the increase in annealing time, as shown in Figure 4 and Table 2. The polarization curves illustrate that each specimen has a passivation zone, and the passivation zone gradually shifts to the left with the increase in annealing time, resulting in a gradual decrease in corrosion current density. This suggests that a more stable passivation film can be formed in alloys with larger grain sizes, leading to better corrosion resistance [40]. When the annealing time lasts for 60 min, the corrosion resistance of the specimen is the best, with the lowest corrosion current, 113.51 μA·cm^−2^ and the highest polarization resistance, 278.03 Ω·cm^2^.

Based on the comprehensive consideration of the above wear and corrosion test results, the 30-min annealed specimen exhibits the wear rate, 4.146 × 10^−4^ mm^3^·m^−1^·N^−1^; the corrosion current density, 136.88 μA·cm^−2^; and the polarization resistance, 243.26 Ω·cm^2^, showing a good trade-off between wear and corrosion performance. Meanwhile, the literature also reported that the CoCrNiSi_0.3_ MEA annealed at 900 °C showed a reasonable balance between strength and ductility [19]. Therefore, CoCrNiSi_0.3_ MEA annealed at 900 °C for 30 min is selected as the substrate for further deposition of CrN films in this study.

### 3.3. Characterization of Sputtered CrN Films

Figure 5a–c show the top views of CrN films deposited on CoCrNiSi_0.3_ substrates with a Cr buffer layer, and Figure 5d–f show the cross-sectional views of CrN films deposited on silicon substrates, with sputtering times of 2 h, 2.5 h and 3 h, respectively. According to these SEM images, all the sputtered CrN films have been successfully deposited on the CoCrNiSi_0.3_ (with Cr buffer layer) and silicon substrates with excellent adhesion and exhibit the same island growth model and columnar grain structure as reported in the literature [41]. According to the cross-sectional SEM images, the width of the columnar grains of the CrN film decreases with the increase in sputtering time. Meanwhile, when the sputtering time is set to 2, 2.5, and 3 h, the film thickness of CrN film increases from 3.83 to 5.02 to 6.40 μm. However, for different deposition times, the deposition rate remains constant at about 2.02 μm/h. 

In this study, low-incidence-angle X-ray diffraction is used to measure the CrN films with different deposition times, as shown in Figure 6. Due to the fact that the sputtered CrN film is so thick, almost no signal from the substrate or buffer layer is detected. According to the XRD diffraction peaks, all samples have a series of CrN peaks with cubic structure, including (111), (200), (220), and (311) peaks [42], and almost no other peaks from secondary precipitation or nitride are observed. The lattice constant of the sputtered CrN films is calculated to be 0.414 nm, which is consistent with that reported in the literature [42,43]. Although each diffraction peak of the CrN films with different sputtering times is in the same position, the (200) peak intensity of the CrN films with longer sputtering times is higher. In addition, the full width at half maximum (FWHM) of the diffraction peak of the CrN film with long sputtering time is larger than that of the CrN film with short sputtering time. It is known that the longer the sputtering time is, the thicker the film is. For the three CrN films with different sputtering times, the FWHM of the CrN (111) and (311) peaks has little change, but the FWHM of the CrN (220) peak is positively correlated with the sputtering time. For 2 h-. 2.5 h-, and 3 h- sputtered films, the FWHM of the CrN (220) peak increased sharply from 0.514^o^, 0.518^o^, and 0.564^o^, respectively. It has been reported that the grain size of the deposited film can affect the FWHM value of the diffraction peak [44,45]. Therefore, the increase in FWHM of the CrN (220) peak is ascribed to the refinement of the columnar grains in the thicker films, as shown in Figure 5d–f.

Figure 7 shows the AFM surface morphologies and 3D-stereoscopic images of CrN films with different sputtering times, as well as the corresponding surface roughness (Ra). In Figure 7a, the 2 h-deposited CrN film exhibits a cellular particle structure, whereas the CrN film with a longer deposition time shows a very different angular particle structure on the AFM surface morphology images, as shown in Figure 7b,c. In the cases of 2.5 h- and 3 h-deposited CrN films, the re-arrangement of surface atoms caused by the continuous bombardment of Cr atoms leads to the transformation of the surface growth model from a cellular particle structure to an angular particle structure as a consequence of the competing growth [46,47]. With the increase in CrN film thickness, the competitive growth among the columnar grains and the limited surface atom diffusion result in greater surface undulations [48]. Therefore, with the increase in deposition time, the average surface roughness of 2 h-, 2.5 h-, and 3 h-deposited CrN films increases from 11.7 nm and 15.5 nm to 17.2 nm, respectively.

### 3.4. Wear and Corrosion Resistance of CoCrNiSi_0.3_ MEA with Sputtered CrN Films

Figure 8a shows the plot of friction coefficient and Vickers hardness vs. the sputtering time of CrN film deposited on CoCrNiSi_0.3_ MEA. The data on friction coefficient, hardness, volume loss, and wear rate are summarized in Table 3. CoCrNiSi_0.3_ MEA without CrN film has a friction coefficient of 1.117 and a hardness value of 219 ± 3 Hv. As the sputtering time increases from 2 h to 3 h, the friction coefficient gradually decreases from 0.960 to 0.741, while the hardness gradually increases from 557 ± 62 to 845 ± 15 Hv. The literature related to nitriding treatment has shown that the hardness increases from about 300 Hv in the internal part to about 1200 Hv in the surface part [34]. However, the hardness of HEA decreases sharply with increasing surface depth because the nitrided layer is not uniform and the nitrogen content decreases with increasing surface depth. In this study, the deposited CrN film increased the surface hardness of MEA by about 300 Hv, which is sufficient to significantly improve the wear resistance of MEA. As discussed in Section 3.3, increasing the sputtering time will increase the thickness of CrN film and refine its columnar grains, resulting in higher surface hardness and good resistance to the grinding ball during the wear process. Therefore, the 3 h-sputtered CrN film has a higher surface hardness and a lower friction coefficient. Figure 8b shows the wear volume loss and wear rate of CoCrNiSi_0.3_ MEA with CrN films deposited at different sputtering times. It can be seen in Figure 8b that the wear volume loss and wear rate gradually decrease with the increase in sputtering time. However, in comparison with the wear properties of the bare alloy in Table 3, showing the wear volume loss and wear rate of 21.62 × 10^−2^ mm^3^ and 41.46 × 10^−5^ mm^3^·m^− 1^·N^− 1^, respectively, the CrN films deposited on the MEA significantly improve the wear resistance. In this study, the 3 h-sputtered specimen has the best wear performance, i.e., the smallest wear volume loss of 1.102 × 10^−2^ mm^3^ and the lowest wear rate of 2.249 × 10^−5^ mm^3^·m^−1^·N^−1^.

Figure 9 shows the SEM images of the wear track after the wear test. The data for wear track width and wear depth are summarized in Table 4. In Figure 9a, the surface morphology of the bare MEA is severely damaged and produces a lot of debris after the wear test. The wear track width and wear depth are measured to be 524.3 μm and 16.91 μm, respectively. The wear morphology consists of the stretch-induced slight ductile cracks in the wear track [49]. In Figure 9b, the width and depth of the wear tracks of the 2 h-sputtered film are measured to be 380.7 μm and 7.11 μm, respectively, and part of the CrN film on the wear track has been peeled off. This is because a thinner film will not provide enough adhesive strength against the grinding during the wear test [50]. Due to the fact that the thickness of 2 h-sputtered CrN film is very thin, the adhesion of the deposited large particles is poor. Therefore, these large particles are easily removed and leave pits [27], resulting in a high friction coefficient and poor wear resistance. In Figure 9c,d, the wear tracks of the 2.5 h- and 3 h-deposited CrN films have clear boundaries; meanwhile, the width and depth of wear tracks are measured to be 175.2 μm and 3.76 μm, 142.5μm and 2.86 μm, respectively. It can be observed on the SEM images that there is no sudden tear of CrN films during the abrasion process, while the high-speed friction between the grinding ball and the surface of the test piece forms an oxide layer surrounding the wear track [23,49]. Figure 9e shows the friction coefficient curves for the substrate and CrN films for various sputtering times. Generally, in the first half of the wear test, the friction coefficient fluctuates sharply due to the relatively uneven surface caused by the initial abrasion. After long-term friction on the specimen surface, the friction coefficient suddenly increases significantly in the second half of the wear process and then remains at a high level. For the bare substrate specimen, the friction coefficient gradually increases after the number of wear cycles reaches 3000, showing a higher friction coefficient compared with the deposited CrN film specimens, which is reflected on the more seriously damaged surface in Figure 9a. With the increase in deposition time, the rising trend of the friction coefficient curve of CrN film becomes slower. Significantly, the friction coefficient curve shows a smaller height and remains stable at the same level in the 3 h-deposited CrN, which presents the lowest friction coefficient of 0.741. These characteristics also indicate that the 3 h-deposited CrN film has the best wear resistance.

Figure 10 shows the potentiodynamic polarization curves of CoCrNiSi_0.3_ MEA with CrN films deposited for different sputtering times and tested in a 1 M H_2_SO_4_ solution. The corrosion current density (I_corr_), polarization corrosion potential (E_corr_), and polarization impedance (R_p_) are summarized in Table 5. By comparing the polarization curve of the CoCrNiSi_0.3_ substrate without depositing CrN film, the passivation area is obviously narrowed, although the passivation area is still generated in the 2 h-deposited CrN film. When the sputtering time is increased to 2.5 h or 3 h, the passivation area in these thicker CrN films does not form. Meanwhile, the corrosion current density and corrosion polarization resistance have been greatly improved, indicating that the CrN film with greater thickness has a good protection effect. Therefore, 3 h-deposited CrN film exhibits the best corrosion resistance, that is, the lowest corrosion current density of 19.37 μA·cm^–2^ and the highest polarization resistance of 705.85 Ω·cm^2^.

For the 2 h-deposited CrN film, the corrosion solution can easily penetrate the defects on the film surface, reach the substrate, and then form the pinholes during the corrosion process [50,51]. As confirmed by the SEM image in Figure 11a, the corrosion process has penetrated the CrN film and corroded the substrate, forming a passivation film. The polarization curve of the 2 h-deposited film in Figure 10 also shows the passivation effect, a result confirmed by the local pinhole corrosion found on the specimen surface in Figure 11a. In Figure 11b, the 2.5 h-deposited CrN film has no pinhole corrosion or passivation on the specimen surface because of its good protection performance. The CrN film is thick enough to avoid the formation of pinholes. Instead, some small particles appear on the specimen’s surface. Such particles are related to the nodular defect formed by the thermal and structural stresses during the deposition of CrN film. They are stable in corrosive environments and can be retained on the surface of CrN film due to their special characteristics of a simple shape and smooth surface [52]. It is reported that a thicker CrN film can delay the corrosion solution from corroding through the coating [48]. Since the 3 h-deposited CrN film has the thickest thickness, the specimen surface in Figure 11c not only does not form pinholes but also has a few residual nodular defects. Combined with corrosion surface morphologies and potentiodynamic polarization results, it can be proved that the 3 h-deposited CrN film has the best corrosion resistance.

### 3.5. TEM Observation of CrN Films

Figure 12 shows the cross-sectional TEM images of various sputtered CrN films with dense columnar grain structures. The growth direction of columnar grains is perpendicular to the substrate surface, as indicated by the white arrow in Figure 12. The average widths of the columnar grains of CrN films deposited for 2 h, 2.5 h and 3 h are 37.10 ± 7.86, 25.23 ± 6.02, and 23.09 ± 4.88 nm, respectively, measured by TEM cross-sectional images. The columnar grain refinement is also confirmed by the increase in FWHM in the XRD patterns from CrN films with longer sputtering times. Since the TEM specimens are prepared from the surface area of the CrN film, the decrease in the width of the columnar grains indicates that the growth of the CrN film along the vertical direction of the substrate is refined with the increase in sputtering time. According to the literature, in the process of film growth, the defects on the film surface will increase with the increase in deposition time, and these defects can provide nucleation sites for columnar grains and further restrict the movement of grain boundaries to refine columnar grains [53,54]. Meanwhile, with the increase in deposition time, the re-sputtering effect and the competitive growth will change the surface morphology of the film. During the re-sputtering process, the atoms deposited on the film surface would be activated or vaporized by the bombarded atoms, which would influence the growth of the film, resulting in a rougher microstructure [55]. 

The summary results of columnar grain width, friction coefficient, and corrosion polarization resistance of CrN thin films with different deposition times are plotted in Figure 12d. Clearly, columnar grain refinement of CrN film with longer sputtering times improves the wear and corrosion resistance. For ceramic films, a reduction in columnar grain size tends to improve corrosion resistance because the electrical resistivity at the columnar grain boundaries is higher than that inside the columnar grains, and even the carrier density at the columnar grain boundaries is lower [56,57]. Meanwhile, columnar grain refinement leads to an increase in the total area of columnar grain boundaries, thereby increasing the total resistance of the film. However, the longer the sputtering time, the thicker the film thickness, which means that the longer the length of the columnar grains, and the refinement reduces the cross-sectional area of columnar grains.

These long and thin columnar grains will increase the resistivity of carrier transport, thereby increasing the Rp value and improving corrosion resistance. Meanwhile, with the increased deposition time, the columnar grain refinement of CrN film will also improve the surface hardness, reduce the friction coefficient, and improve the wear resistance. 

## 4. Conclusions

A series of CrN films with different thicknesses have been successfully deposited on CoCrNiSi_0.3_ MEA with a Cr buffer layer by a DC magnetron sputtering system. Experimental results show that the CrN films remarkably influence the mechanical properties and corrosion resistance of the CoCrNiSi_0.3_ MEA. The important conclusions are as follows:The annealed CoCrNiSi_0.3_ MEA exhibits a single-phase FCC structure, and plenty of annealed twins are also observed in the grains. The grain refinement of CoCrNiSi_0.3_ MEA improves the hardness and wear resistance but decreases the corrosion resistance.Through the depositing of the Cr buffer layer, a super-hard CrN film is successfully deposited on the soft CoCrNiSi_0.3_ substrate. The deposited CrN films exhibit a columnar grain structure. With the increase in deposition time, the width of columnar grains decreases and the average surface roughness of CrN films increases.With the increase of sputtering time, the increase of CrN film thickness, and the refinement of columnar grains, both wear and corrosion resistance improve simultaneously.In this study, CoCrNiSi_0.3_ MEA with 3 h-sputtered CrN film has the lowest wear rate of 2.249 × 10^−5^ mm^3^·m^−1^·N^−1^, and the best corrosion resistance of I_corr_ 19.37 μA·cm^–2^ and R_p_ 705.85 Ω·cm^2^.

## Figures and Tables

**Figure 1 materials-16-01482-f001:**
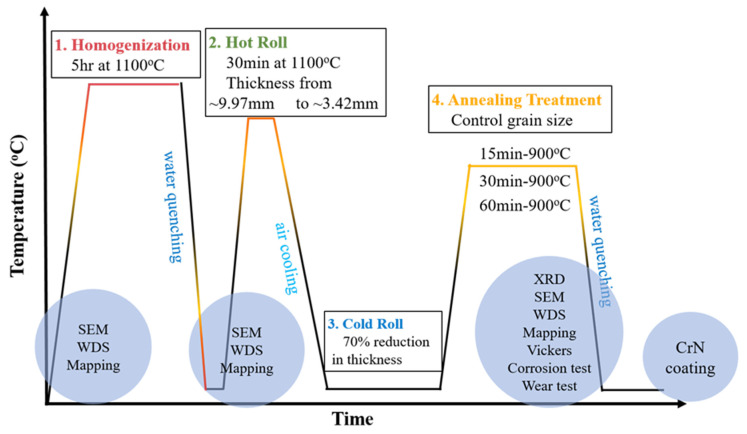
Experimental flow chart of this study.

**Figure 2 materials-16-01482-f002:**
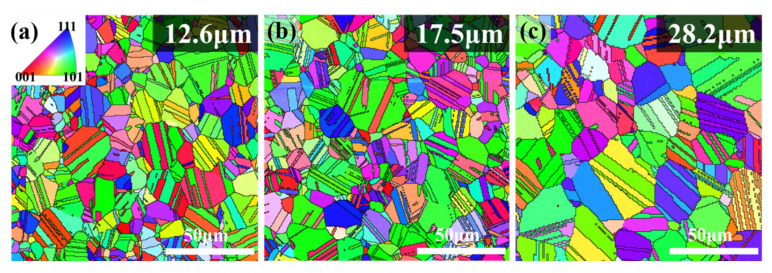
EBSD IPFz maps of the 900 °C-annealed CoCrNiSi_0.3_ MEA. (**a**) 15-min, (**b**) 30-min, (**c**) 60-min.

**Figure 3 materials-16-01482-f003:**
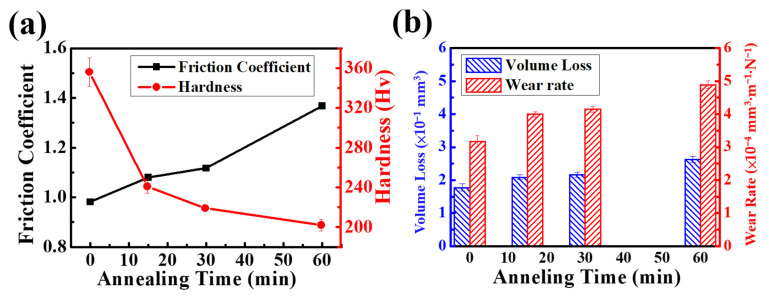
Plots for (**a**) friction coefficient and hardness, (**b**) volume loss and wear rate of CoCrNiSi_0.3_ MEA under different annealing time.

**Figure 4 materials-16-01482-f004:**
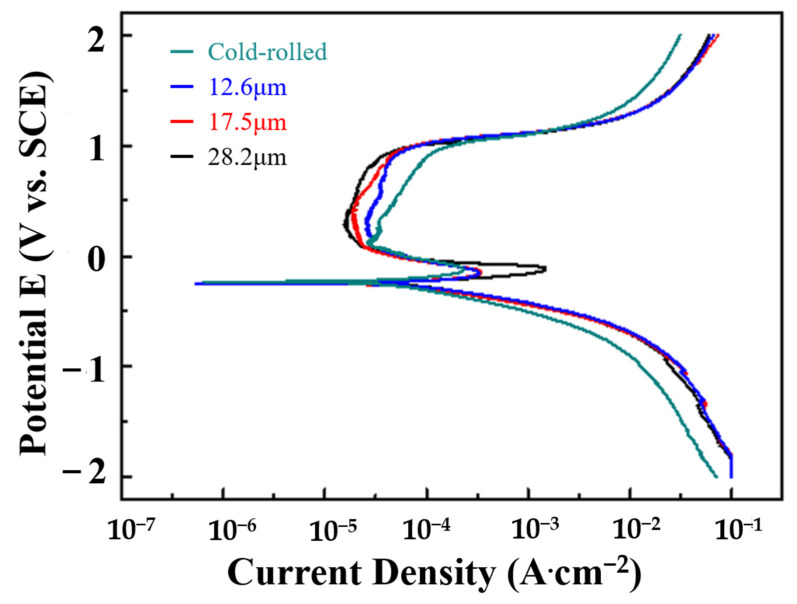
Polarization curves of CoCrNiSi_0.3_ MEA with CrN films deposited for different sputtering times, tested in a 1 M H_2_SO_4_ solution.

**Figure 5 materials-16-01482-f005:**
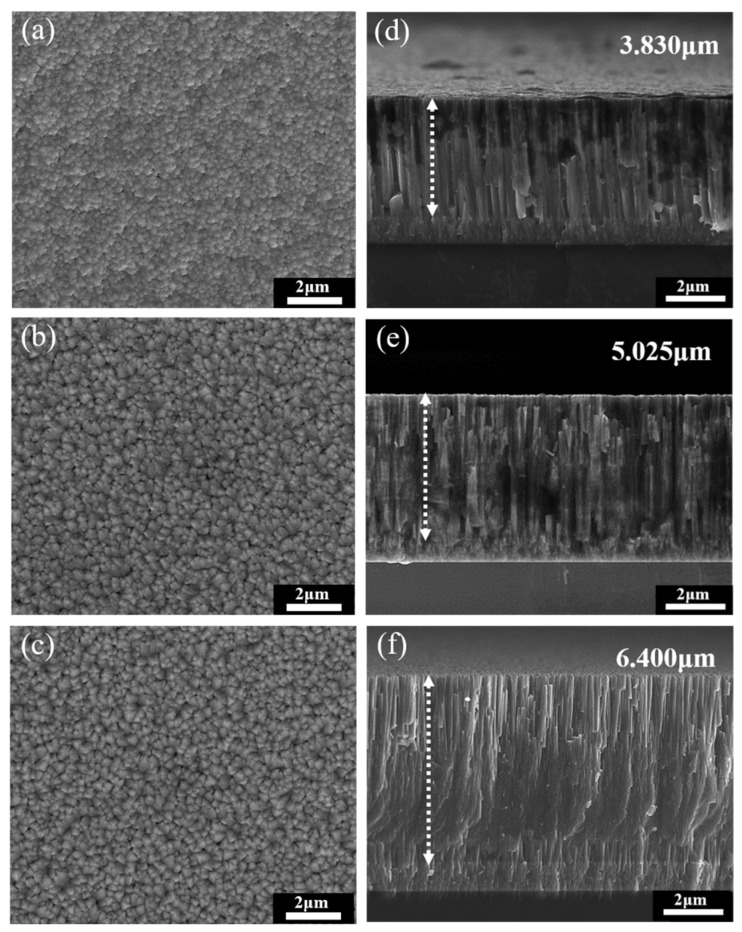
SEM morphologies of (**a**–**c**) over-view and (**d–f**) cross-section of the CrN thin films deposited for (**a**,**d**) 2 h, (**b**,**e**) 2.5 h, and (**c**,**f**) 3 h, respectively.

**Figure 6 materials-16-01482-f006:**
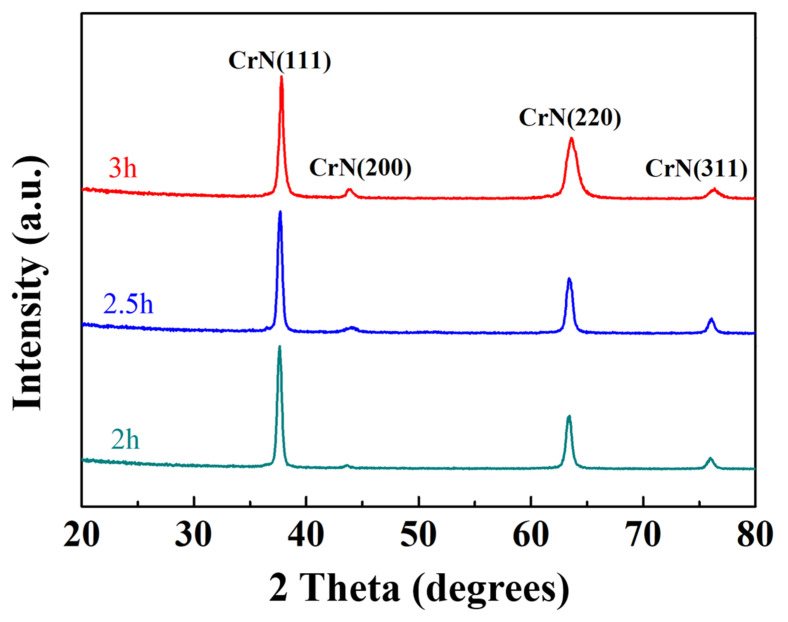
XRD patterns of CrN films deposited under different sputtering time.

**Figure 7 materials-16-01482-f007:**
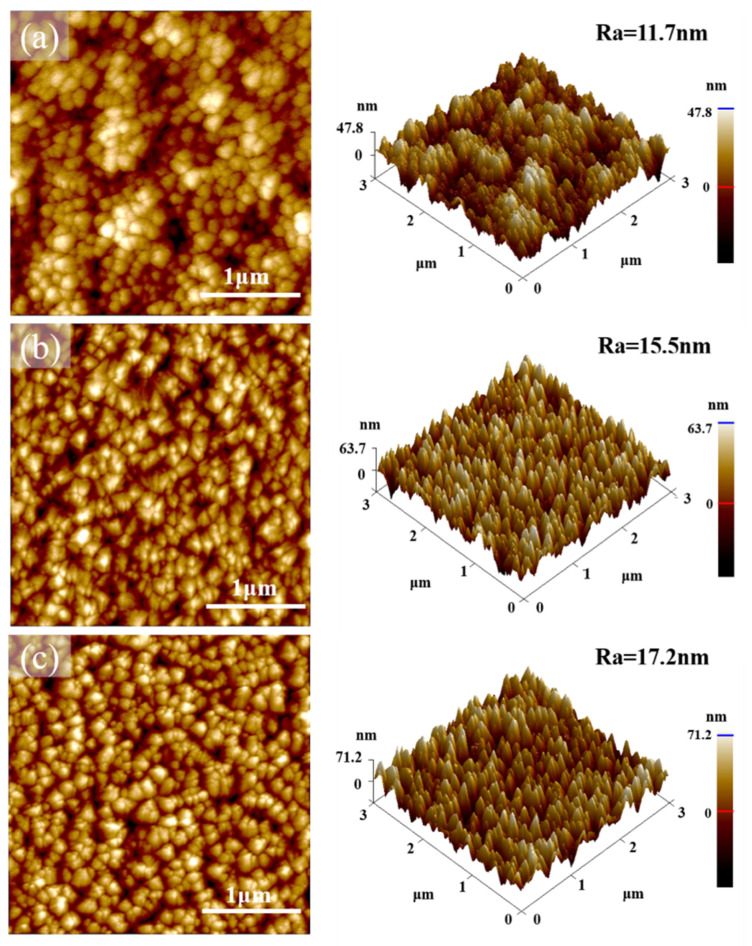
AFM analysis for topographic images of CrN films deposited of (**a**) 2 h, (**b**) 2.5 h, and (**c**) 3 h, respectively.

**Figure 8 materials-16-01482-f008:**
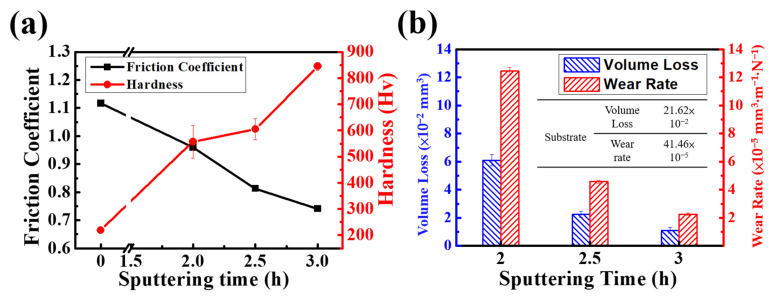
Comparison of the (**a**) Friction Coefficient, Hardness and (**b**) Volume Loss and Wear Rate of CrN films with different sputtering time.

**Figure 9 materials-16-01482-f009:**
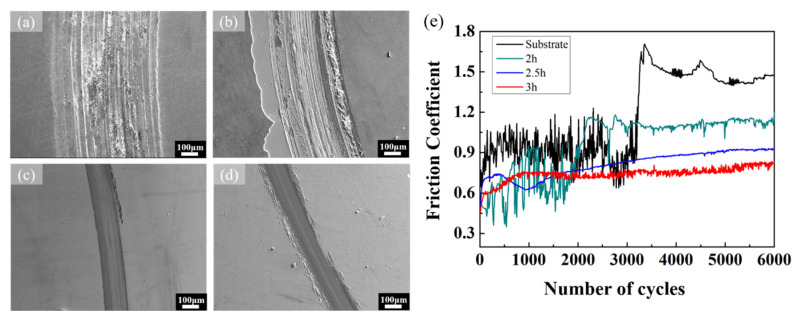
SEM observation for over-view of the wear track (sliding speed 0.2 m·s^−1^ and load 2 N) on MEA substrate with (**a**) bared surface and CrN films deposited for (**b**) 2 h, (**c**) 2.5 h, and (**d**) 3 h respectively, and (**e**) Friction coefficient versus the number of cycles at different sputtering times.

**Figure 10 materials-16-01482-f010:**
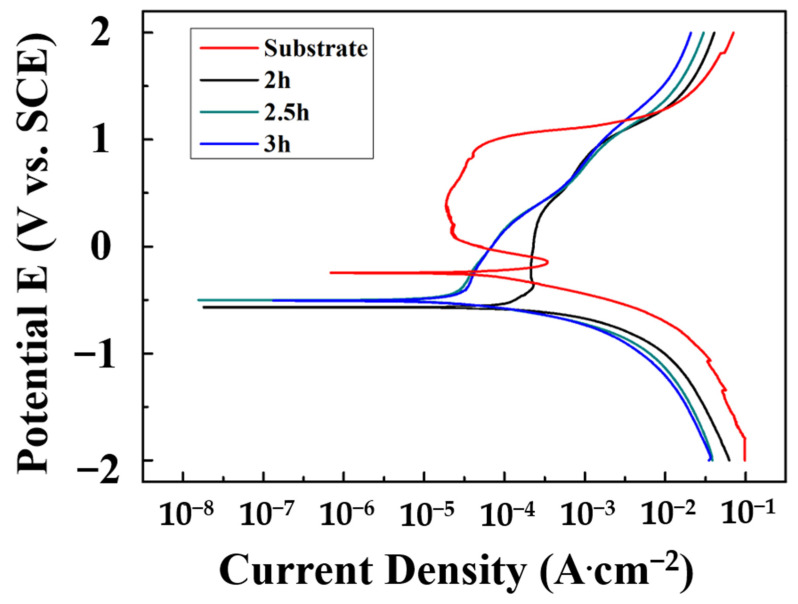
Polarization curves of CrN films sputtered by various time in 1 M H_2_SO_4_ solution.

**Figure 11 materials-16-01482-f011:**
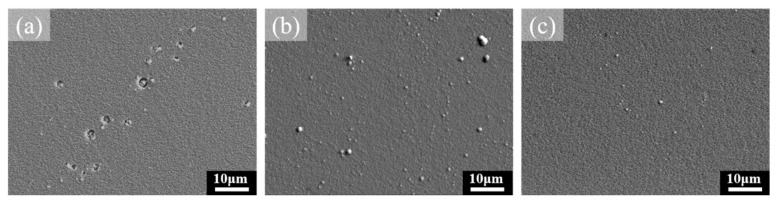
Surface observation of CrN thin films after 1 M H_2_SO_4_ corrosion test deposited for (**a**) 2 h, (**b**) 2.5 h, and (**c**) 3 h.

**Figure 12 materials-16-01482-f012:**
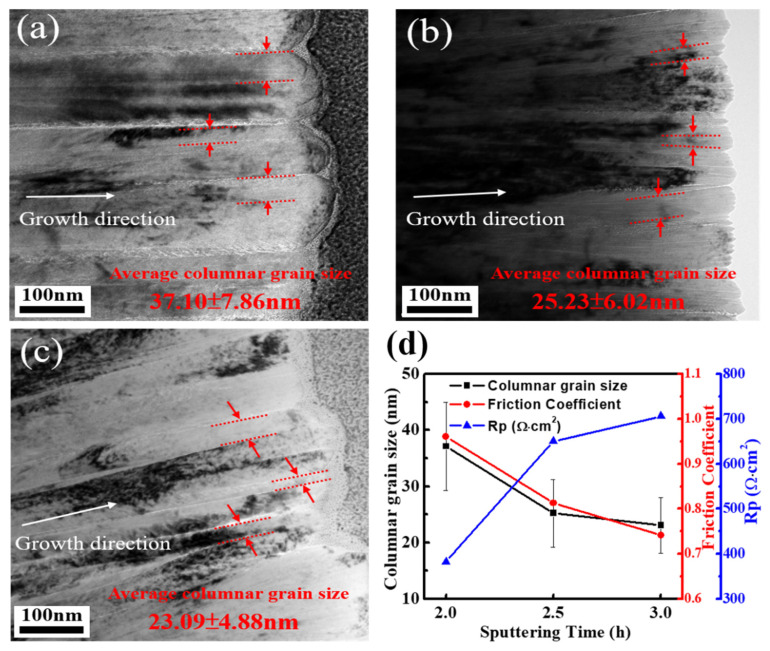
TEM image of CrN thin films deposited of (**a**) 2 h, (**b**) 2.5 h, (**c**) 3 h, and (**d**) plot for columnar grain size, Friction Coefficient and Hardness, and Polarization resistance of CrN films with different coating times.

**Table 1 materials-16-01482-t001:** Friction coefficient, hardness, volume loss, and wear rate of CoCrNiSi_0.3_ MEA with different annealing times.

Annealing Time (min)	Friction Coefficient	Hardness (Hv)	Volume Loss (×10^−1^ mm^3^)	Wear Rate (×10^−4^ mm^3^· m^−1^·N^−1^)
0	0.981	356 ± 14	1.773	3.174
15	1.080	241 ± 7	2.081	3.994
30	1.117	219 ± 3	2.162	4.146
60	1.367	199 ± 6	2.626	4.896

**Table 2 materials-16-01482-t002:** The corrosion current density (I_corr_), polarization corrosion potential (E_corr_) and polarization impedance (R_p_) of CoCrNiSi_0.3_ MEAs with various annealing time tested in a 1 M H_2_SO_4_ solution.

Annealing Time (min)	I_corr_ (μA·cm^−2^)	E_corr_ (mV)	R_p_ (Ω·cm^2^)
0	470.72	−0.230	98.82
15	236.05	−0.223	178.15
30	136.88	−0.214	243.26
60	113.51	−0.213	278.03

**Table 3 materials-16-01482-t003:** Friction coefficient, hardness, volume loss, and wear rate of CoCrNiSi_0.3_ MEA without and with CrN films deposited for different times.

Sputtering Time (h)	Friction Coefficient	Hardness (Hv)	Volume Loss (×10^−2^ mm^3^)	Wear Rate (×10^−5^ mm^3^· m^−1^·N^−1^)
0	1.117	219 ± 3	21.620	41.460
2	0.960	557 ± 62	6.102	12.450
2.5	0.813	605 ± 40	2.245	4.581
3	0.741	845 ± 15	1.102	2.249

**Table 4 materials-16-01482-t004:** Wear track width, and wear depth of CoCrNiSi0.3 MEA without and with CrN films deposited at different times.

Sputtering Time (h)	Wear Track Width (μm)	Wear Depth (μm)
0	524.3	16.91
2	380.7	7.11
2.5	175.2	3.76
3	142.5	2.86

**Table 5 materials-16-01482-t005:** The corrosion current density (I_corr_), polarization corrosion potential (E_corr_) and polarization impedance (R_p_) of CoCrNiSi_0.3_ MEA with CrN films deposited for different sputtering times, tested in a 1 M H_2_SO_4_ solution.

Sputtering Time (h)	I_corr_ (μA·cm^−2^)	E_corr_ (mV)	R_p_ (Ω·cm^2^)
0	136.88	−0.214	243.26
2	103.14	−0.455	402.72
2.5	39.98	−0.420	650.22
3	19.37	−0.415	705.85

## Data Availability

Data will be made available based on a request to the authors.

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
