# Peer review of "Improvement of Corrosion and Wear Resistance of CoCrNiSi0.3 Medium-Entropy Alloy by Sputtering CrN Film"

_materials, 2023, doi:10.3390/ma16041482_

Round 1

Reviewer 1 Report

The authors investigated the tribocorrosion behavior of magnetron sputtering CrN coatings applied to CoCrNiSi0.3 MEA. Although the article contains a novelty, the following points should be taken into consideration.

1-Tribocorrosion is used in the title of the article and when I examine the article, it is seen that the wear and corrosion tests are done separately. Therefore, the term tribocorrosion should be changed to abrasion and corrosion resistance in the title. Because, according to the ASTM40 standard, the wear and corrosion process for tribocorrosion must be done in the same environment.

2- It should be mentioned that after Line 25, the wear resistance of HEA alloys (especially with FCC structure) is not satisfactory due to their low surface hardness, and their wear resistance is improved by methods such as boriding, nitriding and carburizing. Check out the articles below.

a- https://doi.org/10.1016/j.surfcoat.2021.127426

b- https://doi.org/10.1016/j.surfcoat.2010.02.045

c- https://doi.org/10.1016/j.jallcom.2018.07.329

  Otherwise, while HEAs are perceived as perfect alloys, the MEAs you work with may be perceived as materials with low mechanical properties.

3- Give a space in the expressions 1100 ℃ on Line 78.

4- The fonts in Figure 1 are different from each other. All texts in the figure should be corrected as New Times Roman as in the text.

5- With which formulas were the wear volume losses calculated? Give details.

6- Give the properties such as hardness and modulus of elasticity of CrN coatings obtained on the surface of CoCrNiSi0.3 MEA alloys in the result and discussion section. Compare these values with the data obtained from the nitriding of HEAs from the literature.

7- In Figure 9, the wear track widths obtained depending on the different coating times are presented. In addition, the wear resistance can be discussed more clearly by presenting the trace depths.

8- Why were the wear tests of the untreated sample not applied? It is recommended to do it for comparison.

9- Wear traces should be examined with SEM and EDS and the wear mechanisms should be discussed. The way of displaying and discussing the signs of wear can be examined in the accompanying articles.

https://doi.org/10.1016/j.triboint.2021.107160

https://doi.org/10.1016/j.jallcom.2021.161222

10- Line 376“but decreases the corrosion resistance”. The reason can be given briefly.

Author Response

Thank you very much for your comments and suggestions.

Attached is our response to your comments and we have also revised the manuscript according to your suggestions. Please see the attachment.

Reviewer 2 Report

Please mention in the abstract which wear and corrosion properties were considered as reference for the lowest wear rate respectively for the best corrosion resistance?

Line 76, the substantive for “thermomechanical” is missing.

Line 80: why do you mention the various grain-size equiaxed grains for the base alloy? Does it influence the quality of the deposited coating?

Line 89: please mention the hardness of the Cr-buffer layer beside that of the base alloy and of the CrN-coating.

Line 110: please mention the company for the micro-Vickers hardness tester and a wear test machine.

Line 116: please mention the other two electrodes of the three-electrode electrochemical cell.

The information from figure 2 and from point 3.2 are not in accordance with the title of the manuscript. This paper should consider just one quality of the substrate, with the focus on the wear and corrosion behaviour of the deposited coating in comparison with that of the base alloy.

Please adapt the text.

How may samples were tested for evaluation the wear and corrosion behaviour? Please insert the standard deviation for all the hardness values.

It would be interesting to show the evolution of the friction coefficient in time, not only the values.

Please insert in figure 8b also the values for the base material ( as reference) and explain the coating behaviour in comparison with that of the base material.

The first conclusion is not in accordance with the title of the manuscript.

The second conclusion hat to be rephrased, since the adherence of the Cr buffer layer was not tested in this paper.

Round 2

Reviewer 1 Report

The deficiencies mentioned in the previous version of the article have been corrected. The article can be published in its current form.

Reviewer 2 Report

All comments were taken into account.